# Machine learning approaches to enhance diagnosis and staging of patients with MASLD using routinely available clinical information

**Matthew McTeer**[1][◉]*, **Douglas Applegate**[2], **Peter Mesenbrink**[3], **Vlad Ratziu**[4], **Jörn M. Schattenberg**[5], **Elisabetta Bugianesi**[6], **Andreas Geier**[7], **Manuel Romero Gomez**[8], **Jean-Francois Dufour**[9], **Mattias Ekstedt**[10], **Sven Francque**[11], **Hannele Yki-Jarvinen**[12], **Michael Allison**[13], **Luca Valenti**[14], **Luca Miele**[15], **Michael Pavlides**[16], **Jeremy Cobbold**[16], **Georgios Papatheodoridis**[17], **Adriaan G. Holleboom**[18], **Dina Tiniakos**[17,19], **Clifford Brass**[2], **Quentin M. Anstee**[19,20◉], **Paolo Missier**[1◉], on behalf of the LITMUS Consortium investigators¶

1 Newcastle University, Newcastle upon Tyne, United Kingdom, 2 Novartis Institute for Biomedical Research, Cambridge, Massachusetts, United States of America, 3 Novartis Pharmaceuticals, East Hanover, New Jersey, United States of America, 4 Institute of Cardiometabolism and Nutrition, Paris, France, 5 Department of Medicine II, University Medical Center Homburg and Saarland University, Homburg, Germany, 6 University of Torino, Turin, Italy, 7 University Hospital Würzburg, Würzburg, Germany, 8 Servicio Andaluz de Salud, Seville, Spain, 9 University of Bern, Bern, Switzerland, 10 Linköping University, Linköping, Sweden, 11 Antwerp University Hospital, Antwerp, Belgium, 12 University of Helsinki, Helsinki, Finland, 13 University of Cambridge, Cambridge, United Kingdom, 14 Università degli Studi di Milano, Milan, Italy, 15 Università Cattolica del Sacro Cuore, Rome, Italy, 16 University of Oxford, Oxford, United Kingdom, 17 Medical School of National & Kapodistrian University of Athens, Athens, Greece, 18 AMC Amsterdam, Amsterdam, The Netherlands, 19 Translational & Clinical Research Institute, Faculty of Medical Sciences, Newcastle University, Newcastle upon Tyne, United Kingdom, 20 Newcastle NIHR Biomedical Research Centre NUTH NHS Trust, Newcastle upon Tyne, United Kingdom

◉ These authors contributed equally to this work.
¶ Membership of the LITMUS Consortium investigators is listed in the Acknowledgments.
* M.McTeer@newcastle.ac.uk

**Data Availability Statement:** Data underpinning this study are not publicly available. The European NAFLD Registry protocol has been published in [1],

## Abstract

### Aims

Metabolic dysfunction Associated Steatotic Liver Disease (MASLD) outcomes such as MASH (metabolic dysfunction associated steatohepatitis), fibrosis and cirrhosis are ordinarily determined by resource-intensive and invasive biopsies. We aim to show that routine clinical tests offer sufficient information to predict these endpoints.

### Methods

Using the LITMUS Metacohort derived from the European NAFLD Registry, the largest MASLD dataset in Europe, we create three combinations of features which vary in degree of procurement including a 19-variable feature set that are attained through a routine clinical appointment or blood test. This data was used to train predictive models using supervised machine learning (ML) algorithm XGBoost, alongside missing imputation technique MICE and class balancing algorithm SMOTE. Shapley Additive exPlanations (SHAP) were added to determine relative importance for each clinical variable.

including details of sample handing and processing, and the network of recruitment sites. Patient level data will not be made available due to the various constraints imposed by ethics panels across all the different countries from which patients were recruited and the need to maintain patient confidentiality. The point of contact for any enquiries regarding the European NAFLD Registry is the oversight group via email: NAFLD. Registry@newcastle.ac.uk.

**Funding:** This work was supported by Newcastle University and Red Hat UK. This work has been supported by the LITMUS project, which has received funding from the Innovative Medicines Initiative 2 Joint Undertaking under grant agreement No. 777377. This Joint Undertaking receives support from the European Union's Horizon 2020 research and innovation programme and EFPIA. QMA is an NIHR Senior Investigator and is supported by the Newcastle NIHR Biomedical Research Centre. This communication reflects the view of the authors and neither IMI nor the European Union and EFPIA are liable for any use that may be made of the information contained herein.

**Competing interests:** I have read the journal's policy and the authors of this manuscript have the following competing interests: Quentin M. Anstee has received research grant funding from AstraZeneca, Boehringer Ingelheim, and Intercept Pharmaceuticals, Inc.; has served as a consultant on behalf of Newcastle University for Alimentiv, Akero, AstraZeneca, Axcella, 89bio, Boehringer Ingelheim, Bristol Myers Squibb, Galmed, Genfit, Genentech, Gilead, GSK, Hanmi, HistoIndex, Intercept Pharmaceuticals, Inc., Inventiva, Ionis, IQVIA, Janssen, Madrigal, Medpace, Merck, NGM Bio, Novartis, Novo Nordisk, PathAI, Pfizer, Poxel, Resolution Therapeutics, Roche, Ridgeline Therapeutics, RTI, Shionogi, and Terns; has served as a speaker for Fishawack, Integritas Communications, Kenes, Novo Nordisk, Madrigal, Medscape, and Springer Healthcare; and receives royalties from Elsevier Ltd. Jörn M. Schattenberg has served as consultant for Alentis Therapeutics, Astra Zeneca, Apollo Endosurgery, Bayer, Boehringer Ingelheim, Gilead Sciences, GSK, Ipsen, Inventiva Pharma, Madrigal, MSD, Northsea Therapeutics, Novartis, Novo Nordisk, Pfizer, Roche, Sanofi, Siemens Healthineers. Research Funding: Gilead Sciences, Boehringer Ingelheim, Siemens Healthcare GmbH. Stock Options: AGED diagnostics, Hepta Bio. Speaker Honorarium: Advanz, Echosens, MedPublico GmbH. Andreas Geier served as a speaker and consultant for AbbVie, Advanz, Alexion, AstraZeneca, Bayer, BMS,

## Results

Analysing nine biopsy-derived MASLD outcomes of cohort size ranging between 5385 and 6673 subjects, we were able to predict individuals at training set AUCs ranging from 0.719-0.994, including classifying individuals who are At-Risk MASH at an AUC = 0.899. Using two further feature combinations of 26-variables and 35-variables, which included composite scores known to be good indicators for MASLD endpoints and advanced specialist tests, we found predictive performance did not sufficiently improve. We are also able to present local and global explanations for each ML model, offering clinicians interpretability without the expense of worsening predictive performance.

## Conclusions

This study developed a series of ML models of accuracy ranging from 71.9—99.4% using only easily extractable and readily available information in predicting MASLD outcomes which are usually determined through highly invasive means.

## Introduction

Metabolic dysfunction Associated Steatotic Liver Disease (MASLD), formerly known as Non-Alcoholic Fatty Liver Disease (NAFLD) [1] is the world's most common chronic liver disease, and with the rise in increasingly sedentary lifestyles, poses a major challenge to healthcare systems globally. It is estimated that over 25% of the global adult population has MASLD [2], which is predicted to soon be the leading cause of liver transplantation [3]. MASLD encompasses a spectrum of disease severity, ranging from isolated increased hepatic triglyceride content (steatosis; metabolic dysfunction associated steatotic liver—MASL), through hepatic inflammation and hepatocyte injury (metabolic dysfunction associated steatohepatitis—MASH) with increasing fibrosis, and ultimately to cirrhosis and/or hepatocellular carcinoma [4]. More advanced stages of hepatic fibrosis are associated with an increased risk of liver-related and all-cause mortality [5]. The reference standard for grading and staging MASLD is histological using a semi-quantitative scoring system [6, 7]. However, liver biopsy requires expertise in both procurement and histological assessment, are costly, harbour inherent risks and have methodological limitations, (e.g., sampling variability and intra- and inter-pathologist scoring variability), rendering it unsuitable for routine MASLD clinical practice [8, 9]. In recent years there have been major advances in the development of non-invasive biomarkers, both blood-based and radiological [10]. Candidate serum and imaging biomarkers, as well as multi-marker panels, are currently being evaluated in large, multi-centre independent cohorts by international research consortia like LITMUS in Europe and NIMBLE in the USA [11, 12]. However, studies suggest that biomarker performance for the diagnostic context of use remains to date only borderline with classification AUC scores around 0.80 [13]. With no single marker or panel conclusively predicting biopsy results, the hope remains that a combination of complementary assessments may improve diagnostic performance. The application of standard machine learning (ML) approaches to multi-modal training sets remain also relatively unexplored in this research area.

The objective of this study was to investigate the role of selected clinical variables associated with MASL and MASH, when predicting a set of biopsy-derived outcomes that indicate stage of progression along the MASLD spectrum. This work explored the utility of ML approaches

Burgerstein, CSL Behring, Eisai, Falk, Gilead, Heel, Intercept, Ipsen, Merz, MSD, Novartis, Pfizer, Roche, Sanofi-Aventis; received research funding from Intercept, Falk, Novartis. Dina Tiniakos served as consultant on behalf of the University or for ICON, Merck Greece, Madrigal, Inventiva, Histoindex, Cymabay and Clinnovate. This does not alter our adherence to PLOS ONE policies on sharing data and materials. We are not opposed to any reviewers.

to predict binary target conditions in relation to biopsy-derived phenotypes across the MASLD spectrum including At-Risk MASH, Advance Fibrosis and Cirrhosis. Ultimately, we aimed to show that routinely available clinical tests can provide sufficient information to predict these outcomes, suggesting a reduced need to carry out invasive biopsies. Scholars have attempted to tackle this problem through using ML with many studies primarily focusing upon identifying novel combinations of biomarkers that can replace existing surrogate scores that indicate the severity of disease [14–17]. Most studies claim to outperform the existing surrogate markers such as Hepatic Steatosis Index (HSI) and Fatty Liver Index (FLI). The strongest results yielded are from studies that utilise non-routinely collected multi-omics data [18]. Some scholars have focused upon utilising only routinely collected clinical information in their analysis [19–21], however the study cohorts used or the results their method's yielded have been limited.

This paper demonstrates how we achieved our aim through using only data that is easily and readily available from routine clinical appointments and standard blood tests to accurately predict individuals who are at risk of MASH and other outcomes in relation to MASLD severity via ML. We also show that the introduction of variables that are more difficult to obtain into ML classifiers does not improve accuracy significantly to offset the cost of procuring these variables.

## Materials and methods

### Study population

This study utilised data drawn from the LITMUS Metacohort from patients participating in the European NAFLD Registry (NCT04442334), an international cohort of NAFLD patients prospectively recruited following standardized procedures and monitoring; see Hardy and Wonders et al. for details [12]. Patients were required to provide informed consent prior to inclusion. Studies contributing to the Registry were approved by the relevant Ethical Committees in the participating countries and conform to the guidelines of the Declaration of Helsinki. The Metacohort enrolled subjects from sites in Belgium, Finland, France, Germany, Italy, the Netherlands, Spain, Sweden, Switzerland, and the UK between Jan 6, 2010, and Dec 29, 2017. Subjects were at least 18 years old, clinically suspected of having MASLD having been referred for further investigation due to abnormal biochemical liver tests and/or radiological evidence of steatosis. Participating subjects also received a liver biopsy confirming their MASLD status within 6 months of enrolment. After providing written informed consent, participants underwent standardised assessment protocols, including collection of serum blood samples for later analysis with novel biomarkers. Participants reporting excessive alcohol consumption (>20/30g per day for women/men) in the preceding 6 months and/or history of excessive alcohol consumption in the past 5 years were excluded along with participants reporting other causes of chronic liver diseases. Summary statistics of the LITMUS Metacohort at baseline assessment are illustrated in the table S1 Table in the S1 File).

### Features and responses

The branch of ML that this paper focuses on is known as supervised learning classification. This is simply where ML algorithms learn from observations that have been labelled, in this case as either negative (0) or positive (1) for a particular target condition and uses the information about these individuals to create a model that can predict individuals where their status for the target is unknown (i.e., unlabelled). The information in this case refers to the clinical data that is collected, known as features. In this paper, we use a set of non-invasive clinical and novel biomarkers as our set of predictive features. Clinically derived features were collected by

**Table 1. Clinical variables owing to three feature sets used within this analysis.**

| Feature Set | Clinical Variables |
|---|---|
| Core Features | Age, Gender, BMI, Historic Alcohol Consumption (>5 years ago), Insulin Resistance, Hypertensive, Metabolic Syndrome, eGFR, Dyslipidaemia, ALT, AST, GGT, Platelets, Creatinine, Serum Triglycerides, Albumin, Bilirubin, Obstructive Sleep Apnoea, AST-ALT Ratio |
| Extended Features | *Core Features* + FIB4, NFS, APRI, BARD, Waist-to-hip Ratio, Ferritin, IgA |
| Specialist Features | *Extended Features* + Fibroscan Stiffness, CK18-M30, CK18-M65, Pro-C3, Pro-C6, ELF, ADAPT, FIBC3, ABC3D |

a trained investigator (e.g., weight, BMI, comorbidity information) while standard clinical biochemistry (e.g., LDL, HDL, platelet count, ALT, AST, GGT) was measured at each site's local laboratory. Additional biomarkers available included vibration-controlled transient elastography (VCTE; Fibroscan™, Echosens, Paris, France) to measure liver stiffness; the Enhanced Liver Fibrosis (ELF) test [22, 23], measured on the ADVIA Centaur CP system (Siemens, Munich, Germany); and multiple direct collagen biomarkers, including collagen neo-epitopes Pro-C3, Pro-C4, Pro-C6 [24, 25].

Three different combinations of these features which vary in the difficulty of procurement are used, which are referred to as follows:

- Core features—19 clinical variables that are considered standard measurements that are achieved through a routine clinical appointment or blood test.

- Extended features—26 clinical variables that include the 19 Core features plus 7 other features that are either not difficult to acquire but not collected routinely, or composite scores known to be good indicators of MASLD endpoints.

- Specialist features—35 variables that include the 26 clinical features outlined in the Extended feature set and 9 specialist tests that are rarely procured.

These three feature sets are outlined in Table 1 and are described and evaluated individually in [13]. All three feature sets were applied within the ML modelling to predict 9 binarized target conditions, with the number of individuals that exist within the negative and positive class highlighted in Table 2. These targets are recorded by pathologists from liver biopsies. Biopsy evaluation was performed by expert liver pathologists at the recruiting site. Biopsies, when deemed of sufficient quality and size for clinical diagnosis, were assessed using the NASH Clinical Research Network (NASH CRN) scoring system, where steatosis and lobular

**Table 2. MASLD target condition's class distribution.**

| Target Condition | Definition | Negative (0) | Positive (1) | -/+ Ratio |
|---|---|---|---|---|
| MASL vs. MASH | NAS <4 (0) vs. NAS ≥4 (1) | 2776 | 3132 | 0.9: 1 |
| At-Risk MASH | NAS <4 AND/OR F<2 (0) vs. NAS ≥4 AND F≥2 (1) | 4014 | 2010 | 2.0: 1 |
| High Activity | A <2 (0) vs. A ≥2 (1) | 2426 | 3672 | 0.7: 1 |
| Clinically Significant Fibrosis | F <2 (0) vs. F ≥2 (1) | 3771 | 2532 | 1.5: 1 |
| Advanced Fibrosis (Histology confirmed) | F <3 (0) vs. F ≥3 (1) | 4921 | 1382 | 3.6: 1 |
| Cirrhosis (Histology confirmed) | F <4(0) vs. F ≥4 (1) | 5815 | 488 | 11.1: 1 |
| Advanced Fibrosis (Histology & Clinically confirmed) | F <3 (0) vs. F ≥3 AND clinically cirrhotic cases (1) | 5163 | 1510 | 3.4: 1 |
| Cirrhosis (Histology & Clinically confirmed) | F <4 (0) vs. F ≥4 AND clinically cirrhotic cases (1) | 6009 | 664 | 9.0: 1 |
| At-Risk MASLD | Otherwise (0) vs. $2 \geq F \geq 3$ AND NAS ≥4 (1) (Cirrhotics Excluded) | 3979 | 1406 | 2.8: 1 |

inflammation are scored using a semi-quantitative ordinal score described by integers 0–3 and hepatocyte ballooning is scored 0–2. Together these three scores added provide the composite NAFLD Activity Score (denoted NAS in Table 2), ranging from 0 to 8. Disease Activity Score (denoted A in Table 2), is also a composite score ranging from 0–5, consisting of the level of hepatocyte ballooning and lobular inflammation. Fibrosis stage (denoted F in Table 2) is scored with integers 0–4 [6]. Definitions for each target condition are outlined in Table 2 and they are of particular interest as they indicate whether a patient has progressed to MASH or advanced hepatic fibrosis. As an example, an individual is classified as At-Risk MASH (i.e. positive) if the individual has a NAS score greater of equal to 4 as well as a fibrosis stage greater or equal to 2, and if this is not true then the individual is not At-Risk MASH (i.e. negative).

The output of each combination of feature sets to target conditions are models that represent a function of the inputted clinical variables that can accurately predict one or more of these outcomes.

## Learning approach—XGBoost

Due to data sparsity and varying levels of missingness across these 3 feature sets, we focused our research upon algorithms that did not require a fully imputed dataset, namely XGBoost (eXtreme Gradient Boosting) [26]. XGBoost was a highly suitable algorithm for these binary classification exercises, this is due to its consistent high performance in classification tasks and its ability to treat missing values for features as values themselves meaning that no missing imputation is required. Due to XGBoost's characteristics of being a gradient boosting tree ensemble method, it also takes into account issues surrounding variable multicollinearity, with the ability to consider each variable at a single time for each split and not the correlation that it has with other features. This is particularly important in our models that use the Extended or Specialist feature set, as many features used are composites of others.

## Missing imputation—MICE

In preliminary studies we compared a number of supervised learning algorithms upon predicting the binary target condition for presence/absence of "At-Risk MASH" (NAS ≥4 with Fibrosis ≥2) using all predictive features. The issue with these experiments was that unlike XGBoost the vast majority of algorithms require a totally complete dataset and indeed with the Metacohort being a real-world dataset, there were only 3 out of 35 totally complete clinical features. Missing imputation tool MICE [27] (Multiple Imputation by Chained Equations) was necessary. Having conducted these experiments however it was apparent that XGBoost, despite being a tolerant algorithm towards missing values, was still the best performing learning algorithm and when using XGBoost, performing MICE imputation on the training set offered marginal improvements compared to using the algorithm without imputation. Depending upon the data type for each feature, MICE will use different methods to determine the missing value, for example using predictive mean matching for continuous numerical data, logistic regression for dichotomous variables and polytomous regression for categorical data. The level of missing imputation required for 'Core' features is far less than that of the more difficult to procure feature sets, however even of the 19 'Core' clinical features, only 10 were more than 90% complete.

## Class balancing—SMOTE

In ML classification models, imbalanced datasets can result in the model skewing predictions towards the majority class in order to maximise model accuracy. We therefore have the option of either downsampling (removing datapoints of the majority class) or upsampling (increasing

the number of datapoints of the minority class). Downsampling is less favourable due to the removal of perfectly valuable datapoints therefore we focus upon upsampling techniques. SMOTE (Synthetic Minority Oversampling Technique) is an upsampling method that synthetically creates new minority class datapoints through selecting examples of the minority class that are near neighbours to each other in feature space before drawing a hypothetical line between these points; SMOTE then creates a new datapoint at some point along this line thus creating a new synthetic minority class sample [28]. SMOTE is considered to be more reliable than other upsampling techniques due to interpolating new data between existing minority class datapoints, rather than some techniques that simply duplicate existing datapoints which can lead to overfitting. It is important to note however that SMOTE is only operational when there are no missing values across the feature set, therefore all models that use SMOTE must also use MICE.

## Model interpretability—SHAP

Particularly in ML studies for medical domains, there has long existed a tension between the level of understanding of how these classifiers create their conclusions and the overall model accuracy [29]. Medical settings in particular rely upon model interpretability to reduce the level of complexity of a model and therefore increase the trustworthiness of its results. Using an additive feature attribution method known as Shapley values [30], it is possible to determine the relative importance for each feature and has helped to aid the explanation in individual predictions through feature weightings for every model generated in this paper. Shapley values can be added to a model post-hoc to its creation; using Python library SHAP, this allows for easy integration of interpretability into naturally difficult to understand black-box models, such as XGBoost.

## Experimental design

Analysing the 9 target conditions outlined in Table 2 and utilising the 3 feature sets of varying degrees of procurement, we applied 3 different model frameworks: XGBoost, XGBoost with MICE, XGBoost with MICE and SMOTE to every dataset and target combination. This in total provided 81 classifiers. We applied a train-test split to each feature and target condition in the proportion of 80% to 20%—the training data is used to tune the hyperparameters of the XGBoost classifiers to derive a ML model. This model was then cross-validated upon the training data to obtain evaluation metrics; the mean average of the AUC, Accuracy, Sensitivity, Specificity and F1 scores across 5 folds were noted. The model was then applied to the test set to derive predicted values and establish a test set AUC value.

## Results

### Machine learning vs. Univariate linear approach

The ML classifiers created used multiple variables in their decision making and were more powerful and effective at predicting these outcomes than each individual feature alone. Fig 1 compares the training set AUC achieved from univariate logistic regression models upon each of the 35 features explored in the analysis and the training set AUC achieved across all ML models created in predicting At-Risk MASH. All 9 ML models outperformed each individual variable when used in isolation. This demonstrates that the predictive power of these ML models is substantially greater than individual variables that have previously been used in predicting various MASLD outcomes. When comparing test set AUC the ML models performed less admirably, however the differences between training and test performance are small and to be

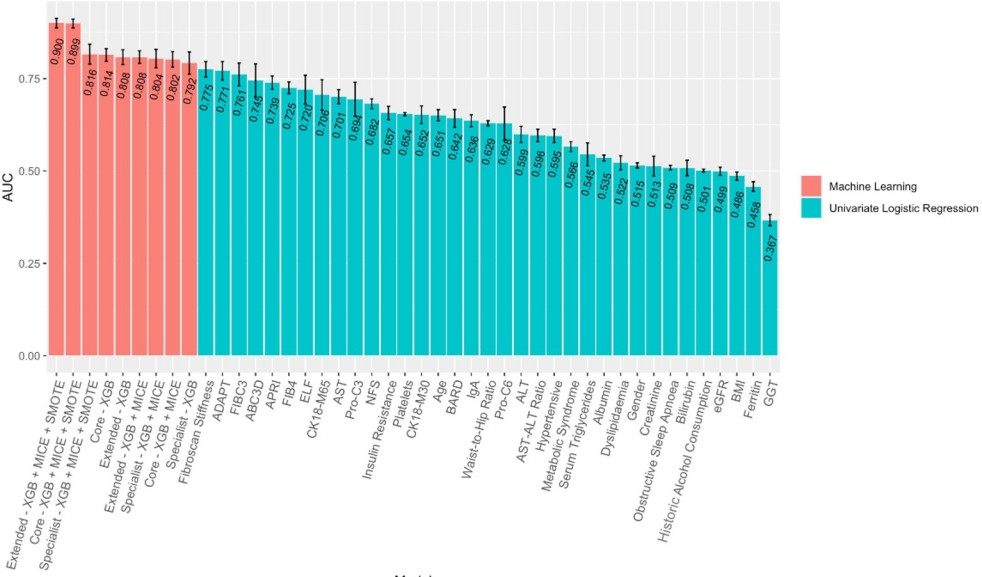

**Fig 1. ML/Linear approach comparison for predicting At-Risk MASH.** Error bars denote +/- S.D from $k = 5$ fold cross-validation.

expected. This is to ensure that our classifiers have not overfitted, that is, where classifiers perform too well upon training data and therefore struggle to generalise to unseen data and predict observations reliably—we discuss our approaches to mitigate overfitting of our ML classifiers in the discussion section of this work. A handful of test set AUC of univariate models can be difficult to compare with our ML classifiers due to the very small sample sizes of some classifiers, such as $N < 200$ in 8 univariate cases, however the majority of these univariate models have test sets comparable to the $N \approx 1200$ of the ML classifiers. Even the smallest sample sizes within the univariate models offer a robust estimator however, as far fewer observations are required in models that utilise only one feature. In general, we see ML models outperforming models that utilise each variable individually, highlighting the progress ML models can offer in improving classifier performance over existing models.

## Modelling using core features

Focusing first on At-Risk MASH as the target condition, the first stage of ML modelling that was undertaken related to whether it was possible to accurately predict individuals with different MASLD outcomes using only Core variables that are routinely collected from either from a routine clinical practice or standard blood test. The Core feature set had 3 combinations of ML modelling applied: XGBoost, XGBoost with MICE, XGBoost with MICE and SMOTE. Each model contained $p = 19$ predictors and had $N = 6024$ observations, of which 2010 were considered 'At-Risk MASH' (positive) and 4014 were not (negative). Taking the 80:20 training-test split, we observed a 2:1 negative-to-positive ratio in terms of the training set class split. We wish only to balance the training set for the XGBoost with MICE and SMOTE model, therefore we artificially enhanced the minority class (in this case the positive set) from 1601 to 3218 to match the case numbers for negative class in the training set. It is important to note that rebalancing was not applied to the test set—this is so the test set is as close as possible to what we would expect to see in reality, thus reducing any model biases.

Good classifier performance was attained using Core variables at predicting At-Risk MASH, with a training AUC of 0.814 for the model which used no imputation or class balancing. There was a markedly improved performance in AUC by ~8% when MICE and SMOTE were used. It is worth noting that the classifier was much better at predicting the negative (and in this case majority) class of individuals who were not At-Risk MASH with average specificity being 86.2%, in comparison to average sensitivity (prediction of positive and in this case minority class) of 63.1%. However, with the use of class balancing algorithm SMOTE the performance of the classifier on the minority positive class improved at the expense of reduced predictive power on the majority negative class, and in fact the classifier accuracy for the positive class was greater than that of the negative class in this case. This was preferable as this ultimately improved all other evaluation metrics of the classifier.

Following the repetition of the 3 combinations of ML modelling techniques upon all other target conditions using the Core variable dataset, as was the case with At-Risk MASH, the best performing classifier was the model that used missing imputation MICE and class balancing algorithm SMOTE. Performance metrics for these models are shown in Table 3.

Strong predictive performance for the XGBoost with MICE and SMOTE classifiers using only Core variables with levels of accuracy was achieved with eight out of nine targets producing an AUC of >0.800. We found that all models that employed SMOTE improved either sensitivity or specificity at the expense of a small decline in the other, providing a more equal result between these metrics than before class balancing was used. For classifiers in which there was a heavy class imbalance such as Advanced Fibrosis (Histology confirmed), Cirrhosis (Histology confirmed), Advanced Fibrosis (Histology & Clinically confirmed), Cirrhosis (Histology & Clinically confirmed) and At-Risk MASLD, the improvement of sensitivity or specificity was greater, and therefore the improvement in overall AUC was greater also.

We also compared the AUC achieved from $k = 5$ training cross-validation with the AUC achieved when applying the models to the test sets to test for overfitting. Cross-validation helps to analyse the generalisation of models upon unseen data, with the main purpose being to provide an estimate of model performance upon new test data. Fig 2 displays the ROC curves for all 5 c.v. folds on the training set with the mean average ROC along with the ROC achieved upon the test set highlighted. The model in question in Fig 2 was the XGBoost with MICE and SMOTE used to predict At-Risk MASH using Core variables only. The AUC for the test set was lower than the mean average for AUC for our training set by approximately 9%. This however was not a significant drop in model performance and with a test AUC performance of 0.80, this displayed good implementation and generalisation of the model upon new and unseen data.

**Table 3. Evaluation metrics for 'Core' dataset performance upon predicting all response using XGBoost with MICE and SMOTE.**

| Response | AUC | Accuracy | Sensitivity | Specificity | F1 |
|---|---|---|---|---|---|
| MASL vs. MASH | 0.719 | 0.663 | 0.658 | 0.667 | 0.661 |
| At-Risk MASH | 0.899 | 0.820 | 0.827 | 0.812 | 0.821 |
| High Activity | 0.801 | 0.723 | 0.720 | 0.734 | 0.724 |
| Clinically Significant Fibrosis | 0.852 | 0.778 | 0.767 | 0.789 | 0.775 |
| Advanced Fibrosis (Histology confirmed) | 0.960 | 0.895 | 0.909 | 0.880 | 0.896 |
| Cirrhosis (Histology confirmed) | **0.994** | **0.964** | **0.980** | **0.949** | **0.965** |
| Advanced Fibrosis (Histology & Clinically confirmed) | 0.961 | 0.901 | 0.915 | 0.888 | 0.903 |
| Cirrhosis (Histology & Clinically confirmed) | 0.993 | 0.960 | 0.973 | 0.947 | 0.960 |
| At-Risk MASLD | 0.921 | 0.846 | 0.856 | 0.835 | 0.847 |

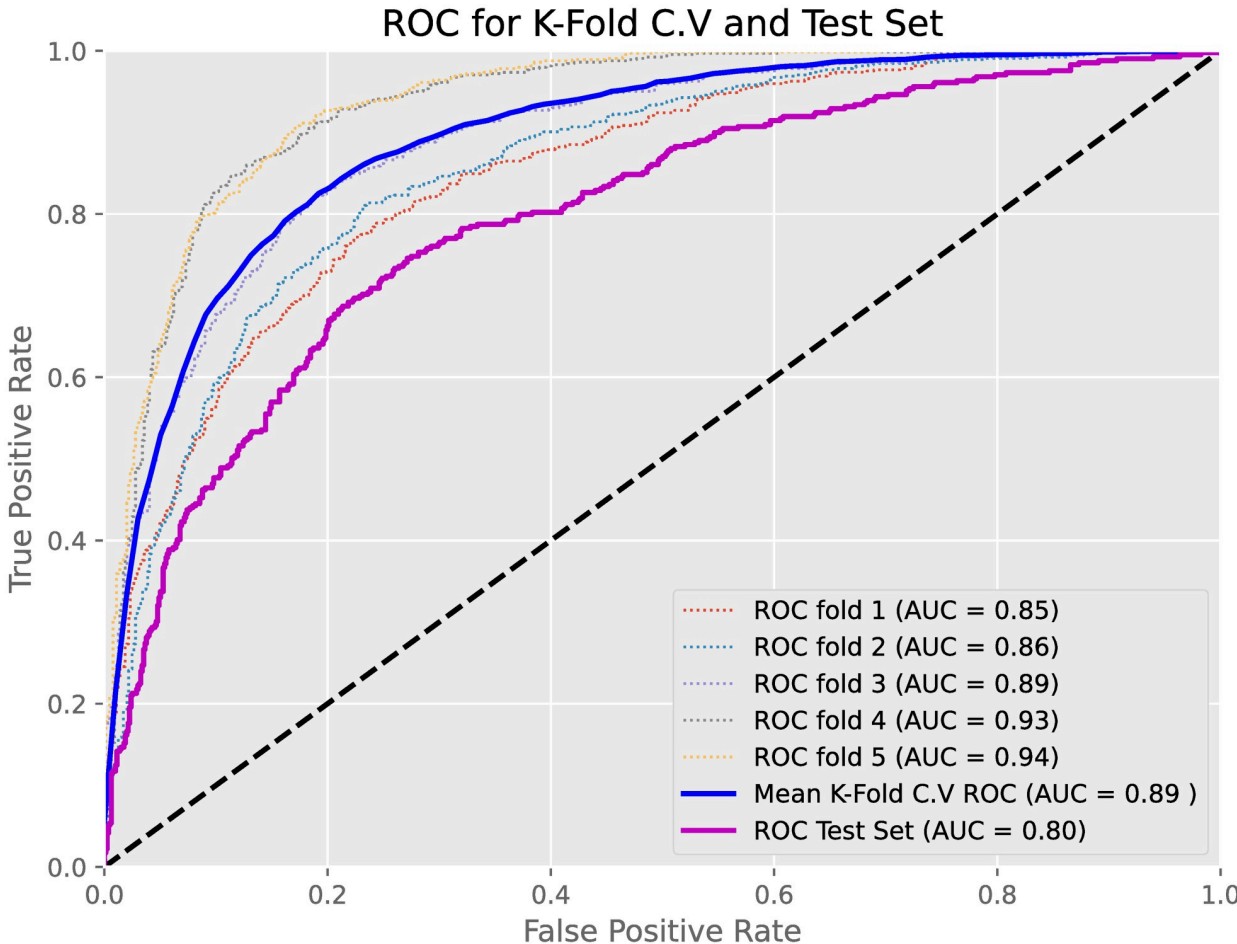

**Fig 2. Training/Test set comparison.** Training and Test AUCs and ROC curves for XGB + MICE + SMOTE model using Core variables upon predicting At-Risk MASH.

## Model interpretability

Applying Shapley values to the most optimal model, we found a clear list of variables in terms of their magnitude on the model's output. Fig 3 ranks the 'Core' features by importance to model prediction from top to bottom, with AST, Platelet Count and AST-ALT Ratio being the most influential predictors of the 19 available. The relative feature value for each variable is presented, with red representing higher respective values for that feature and blue representing lower values. The x-axis of this chart demonstrates the relative 'push' towards positive or negative output of the model—for instance we see that the higher the Age of an individual, the more likely it is that the model will classify an individual as positive and 'At-Risk MASH'.

Interpretability was also available for local predictions as well as the global model. Fig 4 illustrates 4 'force plots' owing to 4 individuals with and without Type 2 Diabetes, and also with and without high stage fibrosis (F >2). Force plots allow visualization of each feature's attribution with 'forces' that either increase or decrease the predicted value of the observation by the model—in the case of the individual without Type 2 Diabetes and low stage fibrosis (top left), strong negative influence from Albumin, GGT and Age outweigh the positive forces from

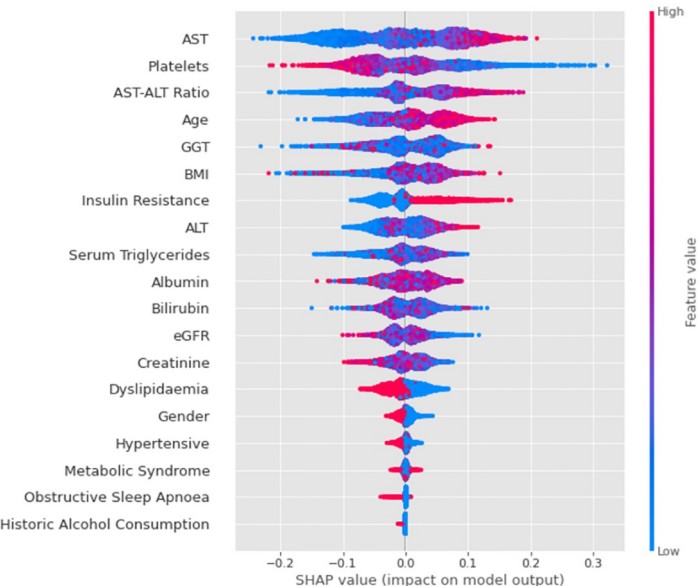

**Fig 3. SHAP summary plots.** Ranking of Core variables in terms of their influence on predicting At-Risk MASH for XGBoost with MICE and SMOTE model.

Insulin Resistance, such that a low prediction output of 0.07 was obtained; this individual would therefore have been predicted by the model to not be At-Risk of MASH.

It is worth noting that an individual's diabetic status and fibrosis stage were not used to train this model, however from the force plots we can see that very high values of 0.99 and 0.96 were predicted for the 2 individuals who were in a high fibrosis stage, therefore considering these individuals to be positively At-Risk MASH. For the 2 individuals who were in a low fibrosis stage, one low prediction value of 0.07 and one high prediction value 0.98 were returned, with the higher value belonging to that of the individual who is diabetic; the model therefore considers the non-diabetic individual with low fibrosis stage to be not At-Risk MASH and the diabetic individual with high fibrosis stage to be At-Risk MASH. All of the force plots have different features that were considered most important in their respective prediction, however common features that appeared in these plots as most critical were Age, AST, and Platelet count.

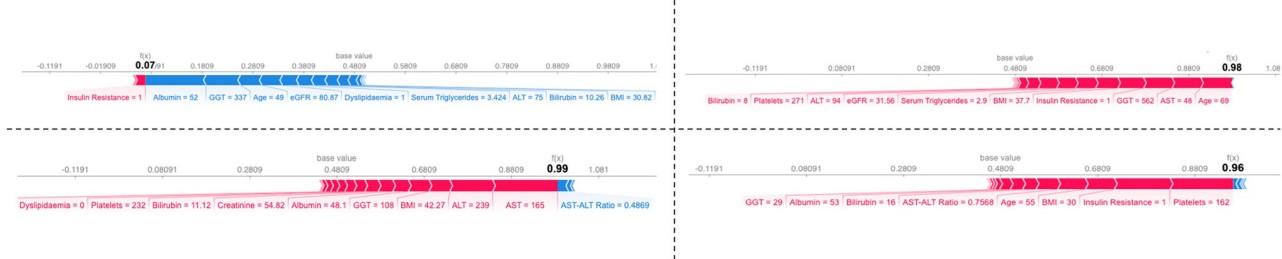

**Fig 4. SHAP force plots.** Force plots illustrating the impact of each feature upon the prediction of 4 random individual's probability of At-Risk MASH. **Top Left**: A non-diabetic, 49 year old man of low fibrosis stage. **Top Right**: A diabetic, 69 year old woman of low fibrosis stage. **Bottom Left**: A non-diabetic 76 year old woman of high fibrosis stage. **Bottom Right**: A diabetic, 55 year old man of high fibrosis stage.

## Modelling using extended and specialist features

Initially focusing again upon the target condition of At-Risk MASH, the analysis was repeated using the Extended and Specialist variables feature set, there were therefore in total 9 models in predicting this response. Fig 5 illustrates the AUC for each of these models.

By directly comparing each model composition to the feature set used, we see that the average improvement in AUC of 0.03% was negligible at predicting At-Risk MASH between Extended and Core feature sets. This compared to an average decline in AUC of -3.4% between Core and Specialist feature sets. When also looking at other performance metrics the improvements/deteriorations were similar for At-Risk MASH. The average improvement in model accuracy, sensitivity, and specificity range between 0.03% and 1.57% when again comparing Extended feature set to Core feature set performance—therefore very little difference was found using the extra 7 variables within this new set of variables. When comparing the change in average performance to the Specialist feature set there was a -4.70% fall in overall accuracy, -8.17% fall in specificity and 6.23% improvement in sensitivity.

This pattern was observed across every target condition explored in this work, with the average improvement of AUC when utilising the Extended feature set over the Core feature set being 0.39%. This was also the case with accuracy, sensitivity and specificity, with the greatest improvement being a 1.17% increase in sensitivity. It seems unlikely the cost of obtaining these extra 7 features offsets any benefit in classification performance.

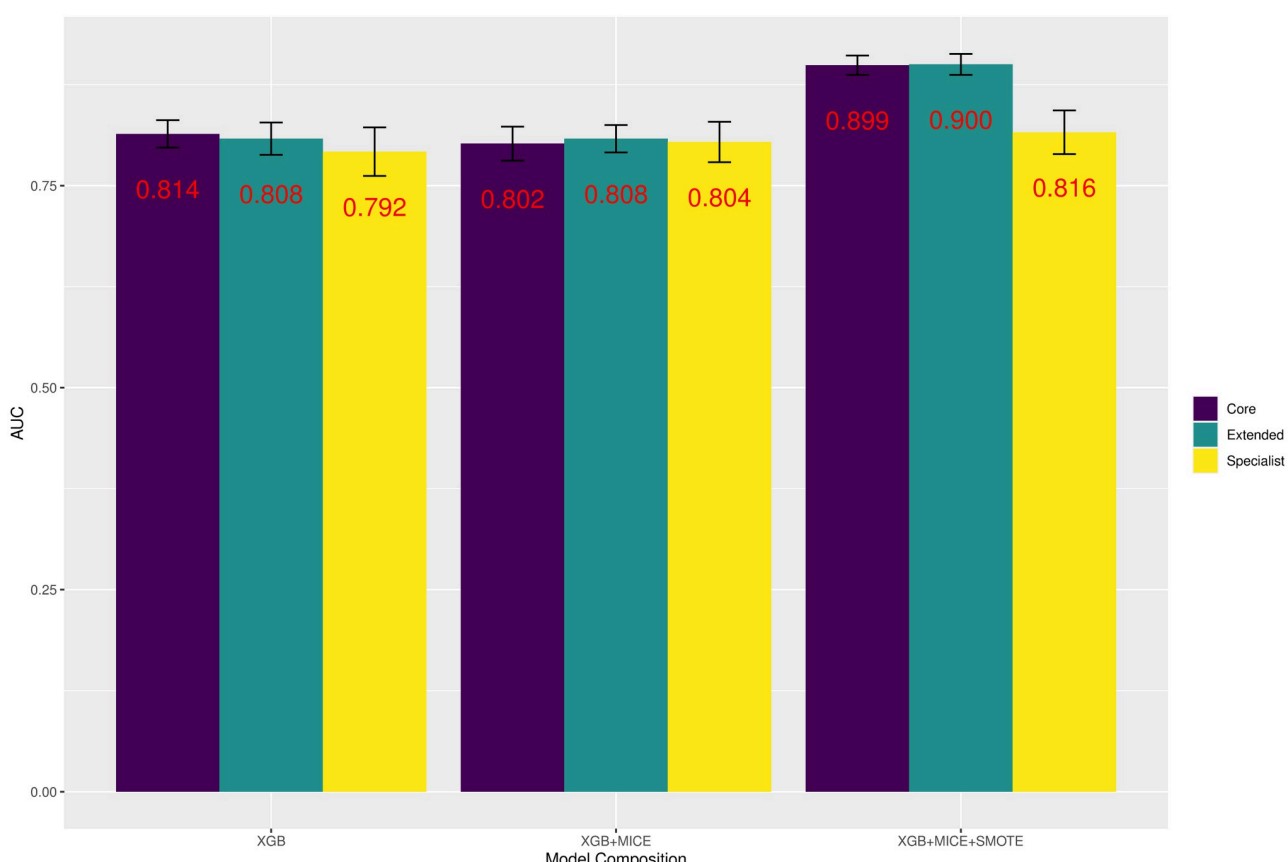

**Fig 5. Modelling for At-Risk MASH.** AUC for every classifier predicting At-Risk MASH by Feature Set and Model Composition.

The differences in average performance metrics between Core and Specialist feature sets were far more variable, however. AUC for predicting MASL vs. MASH and High Activity improved by more than 5% on average, however for other targets such as At-Risk MASH and At-Risk MASLD average AUC performance deteriorated when introducing Specialist features. It is therefore difficult to give a general conclusion for comparing these 2 feature sets in terms of overall performance. Sensitivity however appeared to improve significantly (avg. 10.07%) for every target when the Specialist feature set was used, this typically was at the expense of heavily reduced specificity (avg. -4.86%). The high level of variability between the performance of Core and Specialist feature sets was likely due to the differences in number of individuals available for each set—comparison is therefore more difficult however there is little evidence to suggest that Specialist features perform significantly better than those that can be accessed simply by a routine clinical appointment.

## Discussion

Our best models achieved an AUC score reaching 0.899 in cross-validation and 0.800 in our hold-out test set for predicting At-Risk MASH, and similar performance for other endpoints. These scores largely track the performance observed in [13] and reflect a modest improvement over individual biomarkers. Interestingly, our machine learning models using 'Core' features significantly outperformed established markers such as FIB-4 (with AUC = 0.708 for At-Risk MASH on our test sample). Additionally, they provide similar levels of performance to the best performing specialised markers. This suggests that incremental improvement in MASLD/ MASH screening is possible with established biomarker assays combined with more advanced models. Our inability to radically improve classification performance may be due to the relatively small sample size of novel biomarkers available in the LITMUS Metacohort. Progress with similar models may be possible with the more complete prospective LITMUS Study Cohort [12]. We must also acknowledge that classification performance will be limited by the fundamental variability in biopsy reads, though we do not claim that we have reached that ceiling yet.

It is very typical of medical data to have varying levels of missing data as it is either not feasible or unnecessary to record every variable for each individual at baseline appointments. The level of missingness therefore increases the more complex feature sets are used within the modelling. The level of imputation required for the Specialist feature set was so great for some variables that it is not wise to use the same number of observations as used in the modelling of the other 2 feature sets. Comparison should therefore be treated with caution when comparing classifier performance using Core features to the classifiers using Specialist features with approximately $N \approx 950$ for modelling using Specialist features, compared to $N \approx 6000$ for Core feature modelling.

All target conditions that are explored in this paper are also imbalanced. Ideally for ML classification, the number of negative and positive cases should display a ratio of 1:1, however the nearest that this is achieved for these targets was Steatosis vs. MASH with a ratio of 0.9:1. Some target outcomes display severely imbalanced levels of class, with Cirrhosis (Histology Confirmed) providing a negative-to-positive class ratio of 11.1:1. The usage of class balancing algorithm SMOTE was therefore more aggressive in cases where there are such great imbalance. Also, although SMOTE is a markedly improved version of existing upsampling methods, it naturally still bears limitations. In particular it does not consider the quality of synthetic samples generated and therefore can struggle to fully capture the distribution of the minority class. However, SMOTE still offers significant advantages within ML classification and is seen to be of great benefit and reliability to heavily imbalanced datasets.

The performance for each target condition from base XGBoost model to XGBoost with MICE falls in 8 out of the 9 targets observed suggesting a worse level of classifier performance for the fully imputed datasets. This is in contrast to the use of class balancing algorithm SMOTE, with all model performances for each target condition bar one improving between a range of 5.10–14.60%. For targets in which the level of class imbalancing was more drastic and therefore used SMOTE more aggressively to increase the minority class, it is clear that these targets had the greatest improvements in AUC performance. For target conditions in which the level of initial class imbalancing was less drastic however and the more conservative the use of the class balancing algorithm, there was at worst little difference in classifier performance, however in general there was still an improved model. For all SMOTE models also, any improvement in either sensitivity or specificity was offset by a decline in the other. The only two models in which specificity improves over sensitivity is where the target condition had more positive cases than negative cases. It is useful to point out that for every target condition, the net improvement between these two metrics was greater than zero. Alongside an overall improvement to AUC and Accuracy also, it can therefore be concluded that SMOTE offers a significant improvement upon model performance.

Finally, the reported accuracy of as high as 99.4% in the prediction of Cirrhosis (Histology Confirmed) suggests a possibility of overfitting. Several measures were taken within the experimental design of this work to reduce the chances of model overfit. This included the use of $k$-fold cross-validation when tuning hyperparameters and assessing the model fit; this method allows for the ML models to not be strongly influenced by any particular part of the training data and allow for a more accurate indication of model performance. Furthermore, tuning hyperparameters 'max_depth' and 'colsample_bytree' within the XGBoost models allowed for model complexity to be controlled as well as adding randomness to allow the model training to be robust to noise respectively. SMOTE also ensures that models do not become biased and overfit towards the majority class. It is acknowledged however that despite these steps overfitting can still occur, and as shown in Fig 2, mean training AUC is still slightly greater than test AUC. We would therefore issue caution to clinicians when using these models upon unseen data but be reassured of this small reduction that our models still present overall a good generalisation.

## Conclusion

Building upon previous linear approaches to predict MASLD related endpoints, this research highlights the capability of more complex, non-linear machine learning methods in being able to accurately classify individuals of varying severity in relation to the MASLD natural progression. In particular, we have demonstrated the ability of predicting such outcomes using easily extractable and readily available information as collected from routine clinical appointments or standard blood tests to a high degree of accuracy. Through using the ML algorithm XGBoost along with missing imputation algorithm MICE and class balancing tool SMOTE upon easily accessible variables, we are able to obtain a classifier with an accuracy of 89.9% at predicting At-Risk MASH. Using this model structure, we are also able to accurately predict other MASLD outcomes up to a training set AUC of 99% in some cases. We have also demonstrated that the introduction of variables that are more complex and difficult to obtain from standard healthcare procedures do not substantially improve the accuracy of these classifiers to offset the cost of procuring these variables although confirmatory analysis upon suitable validation sets is required when available. Each model created within this research was also designed to be highly interpretable, offering clinicians the ability to explore how each individual classifier has come to its conclusions. Each model created, with the help of SHAP, was able

to display the most important features used in a model's decision making, how specific values of each feature contribute to final output, and also the observation of personalised predictions for each individual used within the classifier's training.

## Supporting information

**S1 File.**
(DOCX)

## Acknowledgments

The LITMUS consortium, coordinated by Quentin M. Anstee (quentin.anstee@newcastle.ac.uk). Below are all investigators as part of the LITMUS consortium and their respective affiliations:

**Newcastle University:** *Quentin M. Anstee, Ann K. Daly, Simon Cockell, Dina Tiniakos, Pierre Bedossa, Alastair Burt, Fiona Oakley, Heather J. Cordell, Christopher P. Day, Kristy Wonders, Paolo Missier, Matthew McTeer, Luke Vale, Yemi Oluboyede, Matt Breckons.* **AMC Amsterdam:** *Patrick M. Bossuyt, Hadi Zafarmand, Yasaman Vali, Jenny Lee, Max Nieuwdorp, Adriaan G. Holleboom, Athanasios Angelakis, Joanne Verheij.* **Institute of Cardiometabolism And Nutrition:** *Vlad Ratziu, Karine Clément, Rafael Patino-Navarrete, Raluca Pais.* **Hôpital Beaujon, Assistance Publique Hopitaux de Paris:** *Valerie Paradis.* **University Medical Center Mainz:** *Detlef Schuppan, Jörn M. Schattenberg, Rambabu Surabattula, Sudha Myneni, Yong Ook Kim, Beate K. Straub.* **University of Cambridge:** *Toni Vidal-Puig, Michele Vacca, Sergio Rodrigues-Cuenca, Mike Allison, Ioannis Kamzolas, Evangelia Petsalaki, Mark Campbell, Chris J. Lelliott, Susan Davies.* **Örebro University:** *Matej Orešič, Tuulia Hyötyläinen, Aidan McGlinchey.* **Center for Cooperative Research in Biosciences:** *Jose M. Mato, Óscar Millet.* **University of Bern:** *Jean-François Dufour, Annalisa Berzigotti, Mojgan Masoodi, Naomi F. Lange.* **University of Oxford:** *Michael Pavlides, Stephen Harrison, Stefan Neubauer, Jeremy Cobbold, Ferenc Mozes, Salma Akhtar, Seliat Olodo-Atitebi.* **Perspectum:** *Rajarshi Banerjee, Elizabeth Shumbayawonda, Andrea Dennis, Anneli Andersson, Ioan Wigley.* **Servicio Andaluz de Salud, Seville:** *Manuel Romero-Gómez, Emilio Gómez-González, Javier Ampuero, Javier Castell, Rocío Gallego-Durán, Isabel Fernández-Lizaranzu, Rocío Montero-Vallejo.* **Nordic Bioscience:** *Morten Karsdal, Daniel Guldager Kring Rasmussen, Diana Julie Leeming, Antonia Sinisi, Kishwar Musa.* **Integrated Biobank of Luxembourg:** *Estelle Sandt, Manuela Tonini.* **University of Torino:** *Elisabetta Bugianesi, Chiara Rosso, Angelo Armandi.* **Università degli Studi di Firenze:** *Fabio Marra.* **Consiglio Nazionale delle Ricerche:** *Amalia Gastaldelli.* **Università Politecnica delle Marche:** *Gianluca Svegliati.* **University Hospital of Angers:** *Jérôme Boursier* **Antwerp University Hospital:** *Sven Francque, Luisa Vonghia, An Verrijken, Eveline Dirinck, Ann Driessen.* **Linköping University:** *Mattias Ekstedt, Stergios Kechagias.* **University of Helsinki:** *Hannele Yki-Järvinen, Kimmo Porthan, Johanna Arola.* **UMC Utrecht:** *Saskia van Mil.* **Medical School of National & Kapodistrian University of Athens:** *George Papatheodoridis.* **Faculdade de Medicina, Universidade de Lisboa:** *Helena Cortez-Pinto.* **Faculty of Pharmacy, Universidade de Lisboa:** *Cecilia M. P. Rodrigues.* **Università degli Studi di Milano:** *Luca Valenti, Serena Pelusi.* **Università degli Studi di Palermo:** *Salvatore Petta, Grazia Pennisi.* **Università Cattolica del Sacro Cuore:** *Luca Miele, Antonio Liguori.* **University Hospital Würzburg:** *Andreas Geier, Monika Rau.* **RWTH Aachen University Hospital:** *Christian Trautwein, Johanna Reißing.* **University of Nottingham:** *Guruprasad P. Aithal, Susan Francis, Naaventhan Palaniyappan, Christopher Bradley.* **Antaros Medical:** *Paul Hockings, Moritz Schneider.* **National Institute for Health Research, Biomedical Research Centre at**

University Hospitals Birmingham NHS Foundation Trust and the University of Birmingham: *Philip N. Newsome, Stefan Hübscher.* iXscient: *David Wenn.* Genfit: *Jeremy Magnanensi.* Intercept Pharma: Aldo Trylesinski. OWL: *Rebeca Mayo, Cristina Alonso.* Eli Lilly and Company: *Kevin Duffin, James W. Perfield, Yu Chen, Mark L. Hartman.* Pfizer: *Carla Yunis, Theresa Tuthill, Magdalena Alicia Harrington, Melissa Miller, Yan Chen, Euan James McLeod, Trenton Ross, Barbara Bernardo.* Boehringer-Ingelheim: *Corinna Schölch, Judith Ertle, Ramy Younes, Harvey Coxson, Eric Simon.* Somalogic: *Joseph Gogain, Rachel Ostroff, Leigh Alexander, Hannah Biegel.* Novo Nordisk: *Mette Skalshøi Kjær, Lea Mørch Harder, Naba Al-Sari, Sanne Skovgård Veidal, Anouk Oldenburger.* Ellegaard Göttingen Minipigs: *Jens Ellegaard.* Novartis Pharma AG: *Maria-Magdalena Balp, Lori Jennings, Miljen Martic, Jürgen Löffler, Douglas Applegate.* AstraZeneca: *Richard Torstenson, Daniel Lindén.* Echosens: *Céline Fournier-Poizat, Anne Llorca.* Resoundant: *Michael Kalutkiewicz, Kay Pepin, Richard Ehman.* Bristol-Myers Squibb: *Gerald Horan.* HistoIndex: *Gideon Ho, Dean Tai, Elaine Chng, Teng Xiao.* Gilead: *Scott D. Patterson, Andrew Billin.* RTI-HS: *Lynda Doward, James Twiss.* Takeda Pharmaceuticals Company Ltd.: *Paresh Thakker, Zoltan Derdak, Hiroaki Yashiro.* AbbVie: *Henrik Landgren.* Medical University of Graz: *Carolin Lackner.* University of Groningen: *Annette Gouw.* Aristotle University of Thessaloniki: *Prodromos Hytiroglou.* KU Leuven: *Olivier Govaere.* Resolution Therapeutics: *Clifford Brass.*

The code used for the statistical analysis of this work are available in the GitHub repository: https://github.com/mattmcteer/ML-Approaches-To-MASLD.

## Author Contributions

**Conceptualization:** Matthew McTeer, Douglas Applegate, Peter Mesenbrink, Clifford Brass, Quentin M. Anstee, Paolo Missier.

**Data curation:** Matthew McTeer, Vlad Ratziu, Jörn M. Schattenberg, Elisabetta Bugianesi, Andreas Geier, Manuel Romero Gomez, Jean-Francois Dufour, Mattias Ekstedt, Sven Francque, Hannele Yki-Jarvinen, Michael Allison, Luca Valenti, Luca Miele, Michael Pavlides, Jeremy Cobbold, Georgios Papatheodoridis, Adriaan G. Holleboom, Dina Tiniakos, Quentin M. Anstee.

**Formal analysis:** Matthew McTeer, Paolo Missier.

**Funding acquisition:** Quentin M. Anstee.

**Investigation:** Matthew McTeer, Quentin M. Anstee, Paolo Missier.

**Methodology:** Matthew McTeer, Douglas Applegate, Peter Mesenbrink, Clifford Brass, Quentin M. Anstee, Paolo Missier.

**Project administration:** Quentin M. Anstee, Paolo Missier.

**Supervision:** Douglas Applegate, Peter Mesenbrink, Clifford Brass, Quentin M. Anstee, Paolo Missier.

**Validation:** Matthew McTeer.

**Visualization:** Matthew McTeer.

**Writing – original draft:** Matthew McTeer, Quentin M. Anstee, Paolo Missier.

**Writing – review & editing:** Matthew McTeer, Douglas Applegate, Vlad Ratziu, Jörn M. Schattenberg, Elisabetta Bugianesi, Andreas Geier, Manuel Romero Gomez, Jean-Francois Dufour, Mattias Ekstedt, Sven Francque, Hannele Yki-Jarvinen, Michael Allison, Luca

Valenti, Luca Miele, Michael Pavlides, Jeremy Cobbold, Georgios Papatheodoridis, Adriaan G. Holleboom, Dina Tiniakos, Clifford Brass, Quentin M. Anstee, Paolo Missier.

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
