## [Decision Letter · Decision Letter 0]

13 Dec 2023

PONE-D-23-40017Machine Learning Approaches to Enhance Diagnosis and Staging of Patients with MASLD Using Routinely Available Clinical InformationPLOS ONE

Dear Dr. McTeer,

Thank you for submitting your manuscript to PLOS ONE. After careful consideration, we feel that it has merit but does not fully meet PLOS ONE’s publication criteria as it currently stands. Therefore, we invite you to submit a revised version of the manuscript that addresses the points raised during the review process.

We look forward to receiving your revised manuscript.

Kind regards,

Pavel Strnad

Academic Editor

PLOS ONE

Journal Requirements:

Please ensure that your manuscript meets PLOS ONE's style requirements, including those for file naming. The PLOS ONE style templates can be found athttps://journals.plos.org/plosone/s/file?id=wjVg/PLOSOne_formatting_sample_main_body.pdf and
https://journals.plos.org/plosone/s/file?id=ba62/PLOSOne_formatting_sample_title_authors_affiliations.pdf

Thank you for stating the following in the Competing Interests section:

I have read the journal's policy and the authors of this manuscript have the following competing interests:

Quentin M. Anstee has received research grant funding from AstraZeneca, Boehringer Ingelheim, and Intercept Pharmaceuticals, Inc.; has served as a consultant on behalf of Newcastle University for Alimentiv, Akero, AstraZeneca, Axcella, 89bio, Boehringer Ingelheim, Bristol Myers Squibb, Galmed, Genfit, Genentech, Gilead, GSK, Hanmi, HistoIndex, Intercept Pharmaceuticals, Inc., Inventiva, Ionis, IQVIA, Janssen, Madrigal, Medpace, Merck, NGM Bio, Novartis, Novo Nordisk, PathAI, Pfizer, Poxel, Resolution Therapeutics, Roche, Ridgeline Therapeutics, RTI, Shionogi, and Terns; has served as a speaker for Fishawack, Integritas Communications, Kenes, Novo Nordisk, Madrigal, Medscape, and Springer Healthcare; and receives royalties from Elsevier Ltd.

Jörn M. Schattenberg has served as consultant for Alentis Therapeutics, Astra Zeneca, Apollo Endosurgery, Bayer, Boehringer Ingelheim, Gilead Sciences, GSK, Ipsen, Inventiva Pharma, Madrigal, MSD, Northsea Therapeutics, Novartis, Novo Nordisk, Pfizer, Roche, Sanofi, Siemens Healthineers. Research Funding: Gilead Sciences, Boehringer Ingelheim, Siemens Healthcare GmbH. Stock Options: AGED diagnostics, Hepta Bio. Speaker Honorarium: Advanz, Echosens, MedPublico GmbH.

Andreas Geier served as a speaker and consultant for AbbVie, Advanz, Alexion, AstraZeneca, Bayer, BMS, Burgerstein, CSL Behring, Eisai, Falk, Gilead, Heel, Intercept, Ipsen, Merz, MSD, Novartis, Pfizer, Roche, Sanofi-Aventis; received research funding from Intercept, Falk, Novartis.

Dina Tiniakos served as consultant on behalf of the University or for ICON, Merck Greece, Madrigal, Inventiva, Histoindex, Cymabay and Clinnovate.

Please confirm that this does not alter your adherence to all PLOS ONE policies on sharing data and materials, by including the following statement: "This does not alter our adherence toPLOS ONE policies on sharing data and materials.” (as detailed online in our guide for authors http://journals.plos.org/plosone/s/competing-interests).If there are restrictions on sharing of data and/or materials, please state these. Please note that we cannot proceed with consideration of your article until this information has been declared.

One of the noted authors is a group or consortium [insert name of group or team]. In addition to naming the author group, please list the individual authors and affiliations within this group in the acknowledgments section of your manuscript. Please also indicate clearly a lead author for this group along with a contact email address.

Additional Editor Comments :

Reviewers' comments:

Reviewer's Responses to Questions

**Comments to the Author**

1. Is the manuscript technically sound, and do the data support the conclusions?

Reviewer #1: Yes

Reviewer #2: Yes

2. Has the statistical analysis been performed appropriately and rigorously? 

Reviewer #1: Yes

Reviewer #2: Yes

3. Have the authors made all data underlying the findings in their manuscript fully available?

Reviewer #1: Yes

Reviewer #2: Yes

4. Is the manuscript presented in an intelligible fashion and written in standard English?

Reviewer #1: Yes

Reviewer #2: Yes

5. Review Comments to the Author

Reviewer #1: In the manuscript, entitled “Machine Learning Approaches to Enhance Diagnosis and Staging of Patients with MASLD Using Routinely Available Clinical Information” the authors provide an extensive dataset of machine learning models predicting outcome of MASLD. 1) Subjects for the analyses were drawn from the LITMUS Metacohort (derived from the European NAFLD Registry), which is a very well known/established and extensively characterised patient cohort tremendously increasing the overall value of this particular study. Therefore, the authors employed selected clinical parameters associated with MASLD in combination with histopathologic assessments based on a liver biopsy indicating the different disease stages of MASLD and progression to MASH, fibrosis and ultimately liver cirrhosis. The study data suggest that commonly available clinical variables/tests (i.e. anamnesis, biomarkers, elastrography determined as core, extended or specialist features) provide sufficient information to predict MASLD patient outcomes - potentially reducing the need of more invasive tests such as a liver biopsy.

1) “Test set AUC of univariate models and our ML classifiers are also difficult to compare due to the very small sample sizes of some covariates in the univariate modelling, such as N ≈ 150 compared to test set size of all ML classifiers at N ≈ 1200.” I totally agree with the reviewers that it is very difficult to compare both approaches as the total number of individuals/parameters differs enormously. Besides, it is hardly surprising that an unbiased machine learning approach outperforms univariate linear analyses – the additional value of this result is relatively low.

2) “Recalling that we wished only to balance the training set for the XGBoost with MICE and SMOTE model, we artificially enhanced the minority class (in this case the positive set) from 1601 to 3218 to match the case numbers for negative class in the training set. It is also important to note that rebalancing was not applied to the test set - this is so the test set is as close as possible to what we would expect to see in reality, thus reducing any model biases.” Could you please explain to me the reason why you had to increase the number of cases artificially and how does this manipulation do not influence the results. Is the mentioned interpretation of those data reliable? Maybe this should be stressed or at least clarified in the main manuscript for a broad readership.

3) “The average improvement in model accuracy, sensitivity, and specificity range between 0.03% and 1.57% when again comparing Extended feature set to Core feature set performance - therefore very little difference was found using the extra 7 specialist variables within this new set of variables. […] Sensitivity however appeared to improve significantly (avg. 10.07%) for every target when the Specialist feature set was used, this typically was at the expense of heavily reduced specificity (avg. -4.86%).” In conclusion, more commonly available parameters and tests (“core features”) might be more valuable than “specialist features”. This sounds little surprising – yet promising for our daily clinical routine. However, the number of individuals, where all “special features” were available, was significantly lower compared to “core features”. Therefore, one should be very careful interpreting those data and analyses on a greater scale are needed.

4) In reference to “Table 3” highest prediction accuracy was achieved regarding definite parameters / endpoints such as advanced fibrosis or cirrhosis. However, those patients with high inflammatory activity or “at-risk MASH” with advanced or rapidly progressive fibrosis are those patients who needs to be identified to stop further progression to cirrhosis and its complications. In conclusion, one receives the impression that the findings of this machine learning approach are not surprising or novel. Nonetheless, unbiased machine learning approaches will determine the near future of diagnostics and therapeutic interventions improving our daily clinical routines. In this context, the current study provides interesting machine learning approaches with a lack of novelty based on a great database.

Reviewer #2: I congratulate the authors on their timely and interesting manuscript, which focuses on the innovative use of supervised learning in diagnosing the recently renamed Metabolic Associated Steatohepatitis Liver Disease (MASLD).

The study's strength lies in its longitudinal design, encompassing a substantial period from 2010 to 2017, which allows for a comprehensive analysis. The requirement that all participants have biopsy-confirmed MASLD within six months of enrolment adds a significant degree of diagnostic certainty to the study.

The exclusion of participants with excessive alcohol consumption and other chronic liver diseases is appropriate, as it helps maintain the focus on MASLD as the primary condition under study.

The manuscript does an excellent job of demonstrating the application of supervised learning in medical diagnostics. The division of clinical variables into Core, Extended, and Specialist feature sets is a thoughtful approach, offering a layered understanding of data utility in clinical practice.

This manuscript makes a significant contribution to the field of hepatology. The methodological approach is solid, and the insights provided could substantially improve MASLD diagnosis .

Still, I have some comments that need to be adressed:

- The manuscript would benefit from a table 1 describing the baseline characteristics of the cohort and the different subccohorts.

- I wonder why Gender had such a little impact on the results (Figure 3) and how ethnicity was distributed within the cohort.

- I like, that the authors followed the recently published guidelines and Participants reporting excessive alcohol consumption (>20/30g per day for women/men) or other causes of chronic liver diseases were excluded. Why is excessive alcohol consumption in figure 3 even though it was excluded?

- The reported accuracy rate of up to 99.4% suggests a possibility of overfitting. It is essential to discuss measures to mitigate this and how such high accuracy might be interpreted in real-world clinical settings.

- The manuscript would benefit from a deeper analysis explaining the disparity between the excellent performance of individual predictors like AST, Platelet Count, and AST-ALT Ratio in the machine learning models, and the overall poor performance of FIB-4 as a composite score

6. PLOS authors have the option to publish the peer review history of their article (what does this mean?). If published, this will include your full peer review and any attached files.

Reviewer #1: No

Reviewer #2: No

---

## [Author Response · Author response to Decision Letter 0]

1 Feb 2024

Dear Academic Editor and Reviewers,

First of all I, on behalf of all authors of this work, would like to thank you all for your kind comments and constructive feedback of our paper entitled “Machine learning approaches to enhance diagnosis and staging of patients with MASLD using routinely available clinical information”. We would also like to thank you for the opportunity to submit minor revisions to our work taking into account the changes you suggest. In this rebuttal letter we have responded to each of the Editor’s and Reviewers’ comments and highlight where the suggested changes are made within our revised manuscript. 

The Editor and Reviewers’ comments are highlighted here in red and our responses are in black beneath: 

Editor’s Comments:

 1. “Please ensure that your manuscript meets PLOS ONE’s style requirements, including those for file naming.”

A: We confirm that the manuscript meets PLOS ONE’s style requirements as per the template PLOS provides for LaTeX submissions available at https://journals.plos.org/plosone/s/latex. File names have been updated to adhere to these style requirements also. 

 2a. “Please confirm that [Competing Interests] do not alter your adherence to all PLOS ONE policies on sharing data and materials. If there are restrictions on sharing of data and/or materials, please state these.”

A: We confirm the following statement regarding stated Competing Interests: This does not alter our adherence to PLOS ONE policies on sharing data and materials. We have also included this statement within the updated cover letter relating to this submission. 

In response to the editors’ comment regarding data availability, we would like to request an exception, on the grounds that, with reference to the wording below, in this instance public deposition would breach compliance with the protocol approved by our research ethics board. Specifically, the research is based entirely on one of the LITMUS datasets (denoted the ‘1a Metacohort’), which is described in [1]. Such data have been derived from multiple international cohorts, each collected under a separate ethical approval in a different country. At this stage It would be completely unrealistic to seek permission from every different ethics panel to seek permission to share patient level data. We therefore respectfully ask for exemption from the data policy on this occasion. We have also included this response within the cover letter and in the ‘Comments’ section on the online submission. 

[1] Hardy T, Wonders K, Younes R, Aithal GP, Aller R, Allison M, et al. The European NAFLD Registry: a real-world longitudinal cohort study of nonalcoholic fatty liver disease. Contemporary clinical trials. 2020;98:106175. https://doi.org/10.1016/j.cct.2020.106175.

“Data policy:

All PLOS journals now require all data underlying the findings described in their manuscript to be freely available to other researchers, either 1. In a public repository, 2. Within the manuscript itself, or 3. Uploaded as supplementary information. This policy applies to all data except where public deposition would breach compliance with the protocol approved by your research ethics board. If your data cannot be made publicly available for ethical or legal reasons (e.g., public availability would compromise patient privacy), please explain your reasons on resubmission and your exemption request will be escalated for approval.”

 2b. “Please note that exceptions to the data policy are only granted if there are legal or ethical restrictions being placed upon the data by an IRB or ethics committee. At that point we require the authors to provide contact information for an institutional point of contact where fellow researchers can send data inquiries.

If the authors are unable to make the data publicly available, then we ask that the authors provide the source of the data, if it is owned by one or more third parties, or an institutional point of contact, including an email address or phone number, where fellow researchers can send data inquiries.

Please note that PLOS does not allow authors to be the sole contact for data inquiries. If the data is only available upon request, please provide contact information, such as an email address, for a non-author, institutional point of contact (such as an IRB or ethics committee contact) who can field data inquiries from fellow researchers. If the data contact is an individual, please provide their title and relationship to the data as well.”

A: In response to the Editor’s additional revisions from 22nd January and 1st February, we provide the following Data Availability statement: 

Data underpinning this study are not publicly available. The European NAFLD Registry protocol has been published in [1], including details of sample handing and processing, and the network of recruitment sites. Patient level data will not be made available due to the various constraints imposed by ethics panels across all the different countries from which patients were recruited and the need to maintain patient confidentiality. The point of contact for any enquiries regarding the European NAFLD Registry is the oversight group via email: NAFLD.Registry@newcastle.ac.uk. 

Please note that this data contact is not an individual but an institutional point of contact. This statement has also been made clear within the Manuscript in the Acknowledgements section, the Cover Letter and ‘Comments’ on the online submission.

As a courtesy to the Editor, in this response letter we have also included the list of registry sites and PIs as shown in the table below: (see letter)

 3. “Please list the individual authors and affiliations within [group or consortium] in the acknowledgements section of your manuscript. Please also indicate clearly a lead author for this group along with a contact email address.”

A: We have now included all authors and affiliations within the LITMUS Consortium in the acknowledgements section of the manuscript. The lead coordinator of the consortium is Quentin M. Anstee, available at quentin.anstee@newcastle.ac.uk. We have also made this information clear in the acknowledgements section also (page 12). 

Reviewer #1’s Comments: 

 1. ‘“Test set AUC of univariate models and our ML classifiers are also difficult to compare due to the very small sample sizes of some covariates in the univariate modelling, such as N ≈ 150 compared to test set size of all ML classifiers at N ≈ 1200.” I totally agree with the reviewers that it is very difficult to compare both approaches as the total number of individuals/parameters differs enormously. Besides, it is hardly surprising that an unbiased machine learning approach outperforms univariate linear analyses – the additional value of this result is relatively low.’

A: It is perhaps important to stress that only 8 out of 35 univariate logistic regression models had a test set of sample size lower than N=200, and half of the univariate models had a test set sample size N>1000. I have clarified within the manuscript that although comparison may be difficult for a handful of covariates where test set sizes are small, comparison is still useful and more reliable with the vast majority of other covariates used within the univariate logistic regression models. With the univariate models only taking into account one variable each also, far fewer observations are therefore required in order to develop a reliable estimator model with scholars [2,3] (Schmidt, 1971) (Harrell, 2001) arguing that approximately 10-20 observations per covariate are required. Our smallest training sets of N being as low as 150 therefore still allows for the provision of robust models and worthwhile comparisons. Although we agree with the reviewer that the result of ML approaches outperforming univariate linear approaches are unsurprising, we argue that it displays important context in highlighting the progress ML models can offer upon improving classifier performance as opposed to linear models. We have added these arguments within the relevant results section in the revised manuscript (page 7).

[2] Schmidt, F.L., 1971. The relative efficiency of regression and simple unit predictor weights in applied differential psychology. Educational and Psychological Measurement, 31(3), pp.699-714.

[3] Harrell, F.E., 2001. Regression modeling strategies: with applications to linear models, logistic regression, and survival analysis (Vol. 608). New York: springer.

 2. ‘“Recalling that we wished only to balance the training set for the XGBoost with MICE and SMOTE model, we artificially enhanced the minority class (in this case the positive set) from 1601 to 3218 to match the case numbers for negative class in the training set. It is also important to note that rebalancing was not applied to the test set - this is so the test set is as close as possible to what we would expect to see in reality, thus reducing any model biases.” Could you please explain to me the reason why you had to increase the number of cases artificially and how does this manipulation do not influence the results. Is the mentioned interpretation of those data reliable? Maybe this should be stressed or at least clarified in the main manuscript for a broad readership.’

A: In ML classification, an imbalanced dataset can result in the ML model skewing predictions towards the majority class in order to maximise model accuracy. For instance, take the extreme example of a cancer dataset of which 99 out of 100 patients were considered to have a benign tumour and the remaining 1 to have a malignant tumour. The ML classifier, which is trained to maximise the level of accuracy in its predictions, could therefore predict all patients to have benign tumours and receive an accuracy rate of 99%, naturally this is not useful. In an ideal classification dataset, the number of negative-to-positive cases should be 1:1, we therefore have two options: ‘upsample’, i.e. artificially increase the minority class, or ‘downsample’, i.e. remove instances of the majority class. Upsampling is typically preferred to downsampling simply because downsampling involves the removal of perfectly valuable datapoints. SMOTE is an example of upsampling with a focus upon generating new instances of minority class datapoints that can be determined from comparing existing minority class datapoints (interpolation), rather than for example simply replicating minority datapoints exactly. SMOTE therefore is far more reliable in terms of reducing the risk of overfitting which is common in random oversampling techniques. SMOTE naturally has limitations, it does not consider the quality of the synthetic samples generated and therefore may not completely capture the distribution of the minority class, however it is still a markedly improved version of existing upsampling methods and is of great benefit to heavily imbalanced datasets such as those used within this work. I have now stressed in the manuscript why this upsampling took place in the first place (page 6) and discussed the reliability of the interpretation within the discussion section (page 11). 

 3. ‘“The average improvement in model accuracy, sensitivity, and specificity range between 0.03% and 1.57% when again comparing Extended feature set to Core feature set performance - therefore very little difference was found using the extra 7 specialist variables within this new set of variables. […] Sensitivity however appeared to improve significantly (avg. 10.07%) for every target when the Specialist feature set was used, this typically was at the expense of heavily reduced specificity (avg. -4.86%).” In conclusion, more commonly available parameters and tests (“core features”) might be more valuable than “specialist features”. This sounds little surprising – yet promising for our daily clinical routine. However, the number of individuals, where all “special features” were available, was significantly lower compared to “core features”. Therefore, one should be very careful interpreting those data and analyses on a greater scale are needed.’

A: We acknowledge that the results are indeed promising for the daily clinical practice and that impressive levels of ML classifier performance can be achieved through using easily procurable variables, along with the result that more complex to procure variables offer little improvement to classification in our case. We also concur with the reviewer that comparing results between datasets of different sizes requires caution, and that at present the claim that ‘Core’ features are more valuable than ‘Specialist’ features is not yet definitive. We have added that confirmatory analysis is therefore required upon suitable validation sets when they become available within the conclusion section of this paper to make this clear to readers (page 12).

 4. ‘In reference to “Table 3” highest prediction accuracy was achieved regarding definite parameters / endpoints such as advanced fibrosis or cirrhosis. However, those patients with high inflammatory activity or “at-risk MASH” with advanced or rapidly progressive fibrosis are those patients who needs to be identified to stop further progression to cirrhosis and its complications. In conclusion, one receives the impression that the findings of this machine learning approach are not surprising or novel. Nonetheless, unbiased machine learning approaches will determine the near future of diagnostics and therapeutic interventions improving our daily clinical routines. In this context, the current study provides interesting machine learning approaches with a lack of novelty based on a great database.’

A: We concur with the reviewer that the results are perhaps unsurprising however we would argue that ML provides a more advanced form of modelling which can be used in this case as confirmatory of existing clinical dogma, as opposed to being disruptive of expected results from more traditional forms of analysis. We agree also with the reviewer that the LITMUS Metacohort is indeed a great database and is incredibly rich with respect to its field. We would therefore argue that novelty is therefore found through coupling this dataset with robust algorithms based upon state-of-the-art ML to essentially confirm and reinforce something that the clinical practice would expect. 

Reviewer #2’s Comments: 

 1. ‘The manuscript would benefit from a table 1 describing the baseline characteristics of the cohort and the different subccohorts.’

A: For the interest of space within the main manuscript, we have now included a table within the supplementary material to this work summarising the statistics of the LITMUS Metacohort upon baseline assessments. No subcohorts were used or created within this work however since At-Risk MASH is the prevalent response variable within this work, we have also included the summary statistics for individuals who are positive and negative of At-Risk MASH along with the Metacohort as a whole. The characteristics that are displayed within the table include information regarding demographics of the cohort, comorbidities and biomarkers. We have referred to this table on page 3 of the main manuscript. 

 2. ‘I wonder why Gender had such a little impact on the results (Figure 3) and how ethnicity was distributed within the cohort.’

A: Having discussed the results of all models extensively with clinicians within our team there was no medical explanation to why Gender was not considered one of the more significant features. From a ML perspective, one possible explanation is simply the nature of the Gender feature within the dataset being binary and therefore can only ever be 0 (male) or 1 (female). This comparatively offers less information than more continuous, numerical features such as AST, Platelets, Age etc. in which the range of possible values is far greater and therefore there is more data for the XGBoost model to draw conclusions from. We can see however from our SHAP values in Figure 3 that “high values” of Gender (in this case = 1, referring to female) have a more negative impact on the prediction of an individual who is At-Risk MASH, therefore if you are a woman, it is less likely you are to be considered At-Risk MASH, and similarly vice-versa for males. Ethnicity throughout the cohort is almost entirely European Caucasian, with only a handful of individuals from other ethnic backgrounds – for this reason discussion surrounding ethnicity in this work is largely omitted. 

 3. ‘I like, that the authors followed the recently published guidelines and Participants reporting excessive alcohol consumption (>20/30g per day for women/men) or other causes of chronic liver diseases were excluded. Why is excessive alcohol consumption in figure 3 even though it was excluded?’

A: Our apologies to the reviewer, we have edited the manuscript such that on page 3 this now reads “participants reporting excessive alcohol consumption (>20/30g per day for women/men) in the preceding 6 months and/or history of excessive alcohol consumption in the past 5 years were excluded”. The ‘Excessive Alcohol Consumption’ that was used as a feature within the ML models refers to whether or not participants had previous excessive alcohol consumption from over 5 years ago which would therefore not have excluded them from this study. We appreciate this confusion and have therefore changed the original variable named ‘Excessive Alcohol Consumption’ to ‘Historic Alcohol Consumption’ within Table 1, Figure 1 and Figure 3 as well as editing our definition for excessive alcohol consumption as a means of exclusion criteria (page 3). New versions of Figures 1 and 3 have been uploaded. 

 4. ‘The reported accuracy rate of up to 99.4% suggests a possibility of overfitting. It is essential to discuss measures to mitigate this and how such high accuracy might be interpreted in real-world clinical settings.’

A: We agree with the reviewer that accuracy rates as high as 99% do suggest possibility of overfitting of the ML model to the training set. Several steps were taken within the experimental design of this work to reduce the possibility of overfitting including the use of k-fold cross validation upon both hyperparameter tuning and model fitting, the focus upon specific hyperparameters within the XGBoost that are known to control overfitting, as well as the use of class balancing algorithm SMOTE. We have updated the discussion section of the manuscript (page 11) to now showcase our attempts to prevent overfitting within our models as well as discuss their effectiveness. We acknowledge that despite steps being taken to mitigate overfitting, Figure 2 shows that it is still evident through a disparity in mean training AUC and test AUC. In terms of how this result may be interpreted in real-world settings, we would argue that the size of the difference between mean training AUC score and test AUC score is not large enough to cause concern, therefore clinicians when using these models upon unseen data should naturally be cautious about a potential small reduction in performance but can still be reassured of good implementation and generalisation. We have included how the overfitting in our models may be interpreted in a real-world setting within the discussion section of the updated manuscript also (page 11). 

 5. ‘The manuscript would benefit from a deeper analysis explaining the disparity between the excellent performance of individual predictors like AST, Platelet Count, and AST-ALT Ratio in the machine learning models, and the overall poor performance of FIB-4 as a composite score.’

A: We agree with the reviewer’s comment that a deeper analysis would be beneficial, however this is perhaps the most difficult amendment to make to our work. We know with help from SHAP that AST, Platelets and AST-ALT Ratio are the three most important individual predictors in assessing At-Risk MASH when ‘Core’ features are used, however, it is difficult to quantify their importance into a reasonable comparison to the 72.5% training AUC achieved by the univariate logistic regression model using FIB-4 as the sole predictor (as seen in Fig 1). The disparity between the ML and the univariate logistic regression models in general is however one of the key results we would like to promote within this work and on page 7 we have made amendments to highlight the improved performance of using more advanced learning algorithms such as XGBoost over existing forms of analysis such as univariate regression. It is worth noting that FIB-4 in our univariate logistic regression models offers an accuracy score largely in line of what is available in current literature [4] (Vali et al, 2023). FIB-4 is considered to be a good test but not perfect, and exploring its use in large independent cohorts that are well characterised are more likely to give a measure of its true accuracy than the small studies that originally describe its use. In addition to this, generally individual biomarkers are not used in the assessment of MASLD related outcomes but utilised within composite score such as that of FIB-4, NFS and APRI, therefore deeper analysis and wider comparisons across studies of these markers such as AST, Platelet count and AST-ALT Ratio are difficult to make. 

[4] Vali, Y., Lee, J., Boursier, J., Petta, S., Wonders, K., Tiniakos, D., Bedossa, P., Geier, A., Francque, S., Allison, M. and Papatheodoridis, G., 2023. Biomarkers for staging fibrosis and non-alcoholic steatohepatitis in non-alcoholic fatty liver disease (the LITMUS project): a comparative diagnostic accuracy study. The Lancet Gastroenterology & Hepatology.

We would like to once again thank the Academic Editor and Reviewers for their kind comments and constructive feedback for our work, we hope that our responses within this rebuttal letter and our highlighted changes within the new manuscript are sufficient. If there are any further comments you may have surrounding this work we would be more than happy to discuss further. 

Many thanks!

Best Wishes

Matthew McTeer (lead author)

M.McTeer@newcastle.ac.uk

---

## [Decision Letter · Decision Letter 1]

12 Feb 2024

Machine Learning Approaches to Enhance Diagnosis and Staging of Patients with MASLD Using Routinely Available Clinical Information

PONE-D-23-40017R1

Dear Dr. McTeer,

We’re pleased to inform you that your manuscript has been judged scientifically suitable for publication and will be formally accepted for publication once it meets all outstanding technical requirements.

Kind regards,

Pavel Strnad

Academic Editor

PLOS ONE

Additional Editor Comments (optional):

Reviewers' comments:

Reviewer's Responses to Questions

**Comments to the Author**

1. If the authors have adequately addressed your comments raised in a previous round of review and you feel that this manuscript is now acceptable for publication, you may indicate that here to bypass the “Comments to the Author” section, enter your conflict of interest statement in the “Confidential to Editor” section, and submit your "Accept" recommendation.

Reviewer #1: All comments have been addressed

Reviewer #2: All comments have been addressed

2. Is the manuscript technically sound, and do the data support the conclusions?

Reviewer #1: Yes

Reviewer #2: Yes

3. Has the statistical analysis been performed appropriately and rigorously? 

Reviewer #1: Yes

Reviewer #2: Yes

4. Have the authors made all data underlying the findings in their manuscript fully available?

Reviewer #1: Yes

Reviewer #2: No

5. Is the manuscript presented in an intelligible fashion and written in standard English?

Reviewer #1: Yes

Reviewer #2: Yes

6. Review Comments to the Author

Reviewer #1: (No Response)

Reviewer #2: The authors have addressed all my comments sufficiently.

The story is timely and exciting and I was very grateful to serve as a reviewer.

7. PLOS authors have the option to publish the peer review history of their article (what does this mean?). If published, this will include your full peer review and any attached files.

Reviewer #1: No

Reviewer #2: **Yes: **Carolin Victoria Schneider

---

## [Editor Report · Acceptance letter]

16 Feb 2024

PONE-D-23-40017R1 

PLOS ONE

Dear Dr. McTeer, 

I'm pleased to inform you that your manuscript has been deemed suitable for publication in PLOS ONE. Congratulations! Your manuscript is now being handed over to our production team.

Kind regards, 

on behalf of

Dr. Pavel Strnad 

Academic Editor

PLOS ONE